# Promoting Sustainable Lifestyle Habits: “Real Food” and Social Media in Spain

**DOI:** 10.3390/foods11020224

**Published:** 2022-01-14

**Authors:** María Segovia-Villarreal, Isabel María Rosa-Díaz

**Affiliations:** 1Department of Financial Economics and Accounting, Faculty of Business Sciences, University Pablo de Olavide, 41004 Sevilla, Spain; msegvil@upo.es; 2Department of Business Administration and Marketing, Faculty of Economics and Business Sciences, University of Seville, 41018 Sevilla, Spain

**Keywords:** real food, meat food, healthy properties, social media, consumer perceptions, marketing in foods

## Abstract

Obesity and state of being overweight are beginning to be treated as global epidemics. In this context, health professionals are increasingly acting as expert opinion leaders that use social media to connect with the public, in order to promote healthy lifestyles and provide specific recommendations for different product categories, including fresh, processed, and ultra-processed meat products. This study investigates how exposure to content created by health professionals, and posted on social media, influences consumers’ attitudes. For this purpose, the collaboration of one relevant nutritionist influencer in Spain has been obtained. The online survey created has provided 4.584 responses, received from followers (from May to June 2019). After applying a partial least squares path modeling approach, the results suggest that trust in the content shared, the perceived credibility of the professional sharing the information and the informative value, determine the strength with which consumers acquire more knowledge about endorsed products, develop a favorable predisposition towards them, prefer them over their options, and modify their behaviour by purchasing them, instead of their usual foods. The link is stronger, in the case of trust and influencer’s credibility, than for informative value. However, the latter has an indirect effect on the attitude phases through the former.

## 1. Introduction

Chronic diseases, such as cardiovascular pathologies, certain types of cancers, overweight, obesity, and diabetes, are starting to be treated as global epidemics, whose main triggers are unhealthy diets, sedentary lifestyles, and stress [1,2,3]. This is why interest in health consciousness, which refers to the degree to which individuals are concerned with their health, is such a popular topic nowadays [1,4,5].

Previous research claims that several factors, for instance the desire for hedonic pleasure, may guide food choices and contradict the motivation of eating healthy, because they usually lead to less healthy food choices, including ultra-processed meat products. However, this relationship becomes weaker when consumers have greater health consciousness. This is because consumers with high health consciousness engage in more elaborate decision making, by relying on prior health knowledge and beliefs, when evaluating properties of food [3,4]. Nevertheless, consumers find themselves feeling overwhelmed, as a result of the saturation of health consciousness information shared by institutions, health professionals, companies, and the general public [5,6].

Researchers have studied consumers’ perceptions about the reliability of the source of food information and found that they prefer professional and objective sources in this context [5]. However, the content of the information disclosed is not always easily understandable for people who do not master this area of knowledge. At the same time, consumers are also confronted with nutrition and health claims shared by companies in the marketing campaigns of their products, from commercials to packaging. Nutrition and health claims are statements in any message, conveyed in text or image, that establish, suggest, or imply that a food has beneficial nutrition properties [6].

As opposed to information disclosed by scientific and official sources, nutrition and health claims are usually easily comprehensible, thus increasing their chances of attracting consumers’ attention. Consequently, while more consumers are concerned about health consciousness, a rise in the use of nutrition and health advertising messages has occurred in recent years, as an effective marketing strategy to respond to the new demands. Yet, consumers are beginning to distrust nutrition and health claims, due to serious cases of manipulation in food labelling or misleading advertising [6]. Therefore, the need for new strategies to promote healthy eating is clear [3]. These new strategies should focus on consumers’ attitudes towards the associations between food and health in decision making but also considering the aforementioned challenges [3,4]. Taking advantage of the great range of possibilities, offered by the numerous social media (SM) platforms available, many health professionals are creating SM profiles to share their knowledge with the general public [7]. Nutritionists share reliable, professional information online about food items’ properties, regarding health, in a way that is easily understandable for people who are not experts in this area. This has the potential to influence consumer behaviour towards healthy food products.

In Spain, several nutritionists are taking part in this strategy and causing a real impact on their followers’ behaviour and decisions as consumers [8,9,10,11,12,13]. This trend is this movement is known as “Real Food” or “Real Fodding” (initiated in that country by the nutritionist Carlos Ríos, at the beginning of 2017, which promotes the consumption of real food, good-processed food fresh, minimally processed food, or food which industrial or handcrafted processing has not worsened the quality of the natural properties of the specific food) against the consumption of ultra-processed food, which cope supermarket shelves and most occidental diets. This issue takes on special importance in this context, since certain tendencies have been identified among Spanish consumers to associate meat products with unhealthy foods, undervaluing their beneficial health properties. However, at the same time, they are increasing their consumption of ultra-processed meat products.

When this research started, Carlos Ríos had around 500,000 followers on his Instagram account. Now, as the research reaches the end, he has nearly one-and-a-half million. So, it can be stated that the trend is still growing and expected to have more influence over consumers, at the same time as the nutritionists become social media influencers.

The importance of this rising trend can also be observed in the wide variety of articles in newspapers and magazines covering the topic [8,9,10,11,12,13].

For this research, the nutritionist Carlos Ríos has collaborated with us. He promotes healthism, which is the pursuit of specific food choices to enhance physiological function and reduce disease risk. This is achieved by raising health consciousness, through instructional materials about healthy food properties, shared on SM [5]. Due to the wide range of food-related products in supermarkets worldwide, the influencer pairs these suggestions with specific product recommendations. It is to be expected that followers report improvements in their overall health after following the influencer’s guidelines, displayed on his SM accounts, and that they also recommend and advocate for the trend in their environment. Moreover, since the influencer shares useful and hands-on information, regarding specific food products, followers of this trend will take into account the knowledge that they garnered to make more conscious decisions as consumers.

The striking success seems to result from the decoupling from brands’ promotional efforts, the vast references used to base the information provided in influencer-generated content, fact that the person behind the trend is a professional in the health sector who creates visually appealing and easily accessible information content, and possibility of interaction, so that followers can keep in touch with and receive support from the influencer, as well as from other followers. His most popular way of sharing knowledge is usually by going to the main supermarket chains in Spain and explaining the quality of each product by a disclosure of the ingredients, located on the reverse of every food, instead of the nutrition and health claims located on the front of the packaging.

Based on all that has been stated, the present research aims at testing the impact that exposure to content advocating health consciousness may have on consumer behaviour, in terms of awareness, knowledge, liking, preference, purchase intention, and purchase of food-related products linked to the real food content being shared on SM. For doing so, this study addresses preliminary explanations as to how this relationship occurs, by analyzing the role that some variables, such as informative value, trust, and influencer’s credibility, may have in determining the strength of this link. A quantitative study, articulated by an online survey, is selected as a way to assist large data collection. The answers from the 4.584 respondents were analyzed by employing the partial least squares structural equation modeling technique (PLS-SEM), through the SmartPLS 3 computer pack.

## 2. Theoretical Background

### 2.1. Consumer Behaviour, Social Media Influencers, and Health Consciousness

According to consumer behaviour theory, when a strong need is recognized, consumers search for information and evaluate alternatives before making the final decision of performing a purchase, in order to minimize the post-purchase cognitive dissonance that results from inadequate choices [14]. A few decades ago, this process of information search was mainly limited to consumers’ knowledge, collected from their own experience, plus feedback from family, friends, and neighbours, which consumers consider more reliable than companies’ promotional efforts [15]. Nowadays, SM impacts decision making by creating farther connections that allow for the gathering of more trustworthy information and opinions from other participants in social networks [16,17].

Power and Phillips-Wren (2011) defined SM as “web-based and mobile technologies that enhance human communication and create dynamic, interactive dialogues”. When consumers become users of SM, they experience the feeling of being able to know more about products and services, as well as their availability, price, location, and desired attributes [15,18]. This means that they can expand their social environment, and it is usually expected that when this happens, it has an impact on the consumers’ decision process [19].

Traditionally, individuals have taken on different roles, when engaging in interpersonal communications. On the one hand, opinion leaders can influence others’ attitudes and behaviours, using their actions and other interpersonal tools. On the other hand, followers are usually fascinated about opinion leaders and frequently ask them for advice, which will be more likely to be accepted than the advice of non-opinion leaders [19]. With the introduction of SM, users joining social networks started adopting such roles, as would be done in any kind of social relationship [20,21].

Social media influencers are electronic opinion leaders that can increase the influence of the information they receive and transmit to others [21]. According to Freberg et al. (2011) social media influencers “represent a new type of independent third-party endorser who shape audience attitudes through blogs, tweets, and the use of other SM”. In this sense, the influencer-generated content of social media influencers is considered to be more authentic and in direct contact with potential consumers than brand-generated ads [22]. Twitter studies suggest that consumers may accord social media influencers a similar level of trust as they hold for their friends [23].

Usually, social media influencers use their abilities to create visually appealing content, in which a particular product, service, or brand is portrayed, and their characteristics are highlighted in a way that resembles a personal recommendation. Followers of the social media influencers will then have access to all the content posted by the influencer, and the information gathered will affect their consumer decision process to an extent that will depend on variables such as trust on the influencer, age, disposable income, or personal interest [15,20,21].

Consumers usually look for information on SM to get an expert opinion about the massive supply of products that they find in the market, especially in those cases when they simply do not have the background to be able to discern the best products for their needs. In this context, social media influencer- development has shown a professionalization of the sector, which means that experts from different areas of knowledge have created an account on SM platforms to share their expertise. One such example is the rise of health sector-related professionals, such as nurses, gynaecologists, paediatricians, and nutritionists, entering SM to share their knowledge with their followers [7]. This is the case that we aim to explore in this research, especially its connection with consumer behaviour, as a result of gaining information related to health consciousness on SM.

### 2.2. Theoretical Framework and Model

Previous studies already recognized the important role of information garnered through SM in analyzing consumer behaviour nowadays [14,24,25,26,27,28]. Most of the authors studied this link by focusing on the behavioural attitudes of the consumers’ decision process, since they are easily noticed and can be measured. However, from feeling a need to buy something, other complex mechanisms take place in the mind of the consumer, such as liking or preferring the product over other alternatives, that are not considered in those approaches [15]. Moreover, just some of the previous research considers relevant variables that could explain the link between exposure to SM content and behaviour, such as trust in the person sharing the information or information quality [20,21,24].

Based on this literature review, and taking into account the aforementioned purpose of this study, the two most complete frameworks for adopting a holistic approach in the analysis of the exposure to content generated by nutritionists, and its effect on consumer behaviour, could result from the combination of the models used by Duffett (2017) [24], who considers cognitive, affective, and behavioural attitudes, not just behavioural ones, and Lou and Yuan (2018) [20], who takes into account informativeness, credibility, and trust in the content shared, as a way to explain the link.

Duffett (2017) [24] uses the framework of the multicomponent model of attitude phases according to Barry (1987) [29]. The reasons for selecting it in this particular research are twofold. First, it is a suitable way to keep a balance between cognitive, affective, and behavioural responses to SM communications. Previous studies focusing on purchase intentions did not consider the impact that content may have on other stages of the consumers’ decision process, in terms of attitudes [15,18,20,21]. Second, by testing multiple dependent variables, both researchers and companies will be able to identify hidden problems and reaching effects of each relationship for further analysis.

According to Duffett (2017) [24], consumers move through attitude phases: awareness of the brand’s existence (cognitive attitude), knowledge of the brand’s offers (cognitive attitude), liking the brand by having a favourable predisposition to it (affective attitude), displaying preference toward the brand when compared to others (affective attitude), conviction that purchase of the brand would be sensible (behavioural attitude), and the final purchase of the brand (behavioural attitude). These six phases, which are successive and not necessarily equidistant, will be the variables used to measure the effect of content on followers’ behaviours and attitudes as consumers. However, in this research study, instead of measuring the impact on brands, the focus is on products, in general, as a way of assessing the effect of the content on increasing the health consciousness of followers towards healthy food products, in general, as a whole.

Building on this approach to understanding consumers’ attitudes, the underlying mechanisms that explain SM’s effect on consumers were integrated in the model, as a way to provide more understanding of the complete phenomenon. The variables, proposed by Lou and Yuan (2018) [20], serve to provide figures, as to explain some of the determinants of the stated relationship. These variables are influencer-generated content informativeness, as well as trust and influencer’s credibility, measured through the perceived expertise, trustworthiness, attractiveness, and similarity of the influencer-generated content posted on their profile. Lou and Yuan (2018) [20] also suggested that informative value could have an indirect effect over consumer behaviour, through trust and influencer’s credibility.

Based on the literature review and purpose of the study, a model can be depicted to guide this part of the research. The model, shown in Figure 1, portrays the relationship that exposure to real food content on SM, through consumers’ perceived informative value of influencer generated content, as well as trust on it and influencer’s credibility, may have on the different attitudes of the consumer. Additionally, everything that has been stated can be summarized in the following hypotheses:

**Hypothesis** **1** **(H1).**
*Trust in non-branded influencer-generated content on social media and influencer’s credibility positively affects consumers’ attitude phases (cognitive, affective, and behavioural).*


**Hypothesis** **2** **(H2).**
*The informative value of influencer-generated content on social media positively affects consumers’ attitude phases (cognitive, affective, and behavioural).*


**Hypothesis** **3** **(H3).**
*The informative value of influencer-generated content on social media has a positive influence over the degree of trust that consumers place over the influencer-generated content on social media and the influencer’s credibility.*


## 3. Materials and Methods

### 3.1. Case Selection

In order to achieve the objectives of the research, the case of the nutritionist Carlos Ríos, who has succeeded on SM as an influencer, was selected, due to the fact that the percentage of branded content in his profile is extremely low; therefore, the effect of non-branded content could be properly analyzed. This influencer started a trend that became popular in Spain, and other Spanish-speaking regions, by using his expertise and rising influence over other users on SM. Hence, this case seems to fit well with the objectives of this study, namely studying the influence that non-branded, influencer-generated content on SM may have on followers’ attitudes, regarding awareness, knowledge, liking, preference, purchase intention, and purchase of real food and good processed food products. Similarly, once this relationship has been tested, an assessment will be made of the changes in attitudes promoted by the perceived trust in the content generated by the influencer, credibility of the influencer, and informative and entertaining value of the content shared.

### 3.2. Procedure, Measurement, and Sample Selection

An online survey was designed, in line with the objectives of this research and based on the work of some of the authors from the literature review. As explained previously, the main variables of this study are exposure to non-branded influencer-generated content on SM, awareness, knowledge, liking, preference, purchase intention, purchase, informative and entertainment value of influencer-generated content, trust of non-branded influencer-generated content on SM, and influencer’s credibility. To be able to measure all of these variables through the survey method, we used different scales, proposed by Duffett (2017) [24] and Lou and Yuan (2018) [20]. The former included a series of questions in his data collection process, aimed at understanding the link between the multicomponent model of attitude phases, from awareness to purchase, and the exposure to branded, influencer-generated content on SM. In our case, we selected and adapted some of these questions so as to fit our objectives and our specific case. For the measurement of informative and entertainment value of influencer-generated content, trust of non-branded influencer-generated content, and influencer’s credibility, we decided to use the tools proposed in the study of Lou and Yuan (2018) [20].

All of the questions adapted from Lou and Yuan (2018) [20] were anchored by seven-point semantic differential scales, except for the entertainment value, while those from Duffett (2017) [24] were simply formulated using a seven-level Likert scale, which goes from strongly disagree to strongly agree, with the statement proposed in each case [29,30]. They were all displayed following a single positive direction in order to make it easier for the respondent. The use of seven-point semantic differential scales allowed us to capture an acceptable range of variation in the answers, without compromising the simplicity of the survey. Both of these measurement approaches were proven adequate for measuring the variables in previous studies, and the analysis of the measurement model was satisfactory in both of them. The questions included in the survey, as shown to respondents (both in English and in the original language, Spanish), are listed in Appendix A.

Once the objectives and case were selected, and the survey was designed using Google Forms, data collection took place. It started in May 2019 and ended at the beginning of June 2019. In order to carry out a successful and assertive data collection process, we decided to focus on reaching respondents through SM. Moreover, it would not be relevant to consider data about changes in consumers’ decision process of people who did not know the real food trend, because the consumers’ decision process would not be altered at all, due to the lack of new information linked to non-branded, influencer-generated content on SM about the real food. So, we were interested in getting as many answers as possible from people following the real food trend, either through the main social media influencer, Carlos Ríos, or through any other influencer who gave information about it frequently. Therefore, two filter questions were included in the survey, to be able to discern answers from people who knew the movement and whose changes in consumers’ decision process could be studied. Not only that but we also contacted Carlos Ríos to ask him to distribute the survey amongst his followers, in order to reach the objective population. Finally, he agreed, and we were able to incorporate very precise and representative data from the population.

There are many ways in which we could measure the research population, as proposed by previous authors. They stated that the number of followers that an influencer has does not reflect the actual active engagement in the information shared. The fact that people follow an influencer but do not see or interact with the content may mean that they are not getting the information shared in non-branded, influencer-generated content about the real food and, consequently, their consumer behaviour would not be affected. Hence, other indicators should be considered, such as the number of likes in the content posted on the different social media platforms [21,24,31]. Thus, a few figures could be displayed in this study, although we are going to make it simple and share only two figures. A total of 800,000 is the number of followers that Carlos Ríos had in June 2019, although it changes every day, while the number of likes in the content, shared during that same month, oscillates between 13,800 and 31,400. The final sample consisted of a total of 4595 answers, from which 4584 went beyond the filter questions; so, depending on which figure we consider, we would be talking about a 0.57% or between a 14.6% and a 33.2% representation rate.

From these 4584 respondents, apart from their rating of the indicators for each of the nine variables, we obtained information about their age, gender, nationality, level of education, and social strata, to be able to provide a demographic depiction of our sample, which will be displayed in Section 4.1. Moreover, four questions were included at the end of the survey that will serve us for the discussion and implication of results. They were about their perceptions, as active SM users and consumers, of the industry and how well it is adapting to the challenges posed by the real food guidelines, when talking about healthy eating. This was motivated by the fact that on SM, followers of the real food trend are continuously sharing discomfort about going to the supermarket and finding a lack of supply of products that fall within the real food guidelines, which would imply a misunderstanding or disregard of this situation by companies from the sector.

### 3.3. Data Analysis

Once data collection took place, a partial least squares (PLS) path modeling approach was adopted and articulated by employing the SmartPLS 3 computer pack to perform both measurement validation and structural modeling [32]. Moreover, PLS path modeling is also recommended over CB-SEM for testing complex models with many latent variables, as in this case that we have nine latent variables [33,34].

Regarding the sample size of this study, it meets the condition imposed by PLS-SEM of being at least 10 times the greatest number of structural paths predicting a specific construct.

The latent variables in the current model all have reflective measurements (the latent variables precede the indicator). In this manner, the indicators represent effects of the latent constructs and are highly correlated [34]. For this reason, in the use of the selected computer pack for this research, SmartPLS 3, the consistent PLS algorithm and consistent bootstrapping options have been selected. These options perform corrections of reflective constructs’ correlations to make results consistent with a factor-model.

## 4. Results

### 4.1. Socio-Demographic and Consumer-Related Characteristics of the Sample

With respect to the main characteristics observed in our sample, Table 1 provides a comprehensive overview of respondents’ socio-demographic and consumer-related variables included in the survey.

### 4.2. Descriptive Analysis of the Study Variables

The descriptive data of the variables included in our study are shown in Appendix B. Its analysis allows several conclusions to be drawn. First, looking at the awareness phase, the vast majority of respondents agreed with the fact that the content shared by real food influencers made them rethink the way to buy and consume food, and found the content was also useful for being aware of alternatives to ultra-processed food and started paying attention to those. Nevertheless, only 91% of respondents clearly remember the content posted, of which 43% strongly agreed. Overall, followers do become more aware of alternatives to the ultra-processed food, according to the real food trend.

Moving onto the second attitude phase, knowledge, 98% of respondents agreed with the statement, and 70% of them chose the highest level of agreement. This allows us to assert that the knowledge garnered through non-branded influencer-generated contents on SM is being incorporated in the decision-making process of real food followers.

Focusing now on the liking phase, the great majority of respondents begin to value real food and good-processed food more after encountering real food content on SM, at the same time as they started to develop a critical view of ultra-processed food. In this case, the number of respondents choosing the highest level of agreement is remarkable, as 72.6% and 71.4% of respondents, respectively, said that they strongly agreed with the liking-related statements.

Then, talking about developing a positive attitude towards products related to the real food trend, the preference phase would be a step further than liking. In this case, 96% of respondents asserted that the content encouraged them to prefer real food and good-process food, at the same time as they become more reluctant to believe everything that is broadcasted in ultra-processed food’s marketing campaigns. The main reason is that they have the knowledge, derived from non-branded influencer-generated contents on SM, about the real food trend that allows them to discern whether or not these products are actually worth, in terms of health facts, rather than promotional efforts. Around 60% of respondents opted for the highest level of agreement in both preference statements.

Following this line of reasoning, once consumers are aware of a product, know enough about it, and like it over other alternatives, it is only right to think that an intention towards purchasing the product will start emerging in the mind of the consumers. When considering both purchase intention, as well as the actual Purchase of the product, the level of agreement with the statement declines. It may be due to the fact that other variables will come into play (for instance, the level of engagement with making an important change in their diets) and will largely differ from one consumer to another.

Finally, when talking about the last attitude phase, purchase, the tendency observed in purchase intention repeats itself; that is, the percentage of respondents agreeing with the statements proposed is not as high as for the first attitude phases.

In the next sections, we proceed to test some exogenous variables that could help understand this link between non-branded, influencer-generated content on SM about the real food trend and the different attitude phases of consumers who are followers of the trend.

### 4.3. Measurement Model Validation

Measures related to internal consistency and convergent validity needed to assess the measurement model in this research (all constructs are reflective) are included in Table 2. First, most of the items’ loadings are almost equal to or higher than 0.7 (the factor loading for every item should be 0.6 or higher) [35]. When analyzing the average variance extracted AVE (for further testing convergent validity), all constructs have a value higher than the threshold of 0.5, which represents the minimum threshold for AVE. Therefore, we can assert that, overall, the constructs have an acceptable convergent validity level.

Second, when assessing the internal consistency of the measurement model, again, all constructs perform above minimum of 0.7 for composite reliability. Consequently, all the reflective constructs and dimensions seem to be reliable. With respect to discriminant validity (Table 3 and Table 4), according to both criteria, i.e., the Fornell and Larcker’s criteria (1981) and the heterotrait-monotrait ratio of correlations (HTMT), there is evidence of discriminant validity [33].

To measure multicollinearity, VIF values were used. According to the VIF values, all elements range between 1.575 and 2.740. Therefore, no VIF value exceeds the suggested threshold value of 3.0, which implies that we have not encountered collinearity problems [34].

### 4.4. Structural Model and Hypotheses Testing

In order to assess the quality of the structural model proposed in the specified PLS path model, we ran a consistent PLS-SEM algorithm to estimate the model’s path coefficients. Then, we performed a second consistent bootstrapping analysis (5000 resamples) to obtain each path coefficient’s standard deviation and p-value [34]. The results from these analyses are displayed in Table 5 and Table 6. The first one includes all the effects that informative value and trust, combined with influencer’s credibility, have on each of the attitude phases included in this research. The second table shows the indirect effects that informative value has over the attitude phases through trust and influencer’s credibility.

As can be observed, all relationships are positive and significant, with the strongest ones being those between informative value and awareness (0.346), informative value and trust and influencer’s credibility (0.656), trust and influencer’s credibility and awareness (0.378), trust and influencer’s credibility and liking (0.347), trust and influencer’s credibility and purchase (0.368), and trust and influencer’s credibility and purchase intention (0.396). Moreover, the t-statistics’ values disclosed and standard deviations lead us to believe that each of the effects proposed in this research is significant. Likewise, when analyzing the significance of the indirect effects that informative value triggers in the attitude phases, through trust and influencer’s credibility, the takeaways are very similar. In this case, as displayed in Table 6, all path estimates are positive and between 0.2 and 0.3. The *p*-values are also 0, and both the standard deviation and the t-statistics show that the indirect relationships proposed in the research are significant [34].

With respect to the predictive power, in terms of the amount of variance in the endogenous constructs explained by all the exogenous constructs linked to it (R^2^), as well as of the effect sizes (f^2^), results obtained are depicted in Figure 2 and Table 7.

In order to assess the predictive power of the model, the reference has been the evaluative cutoff points (0.25 = weak, 0.50 = moderate, 0.75 = substantial), proposed by Hair et al. (2019) [34]. The results obtained indicate that trust and influencer’s credibility and informative value explain 43.4% of variance in awareness, 32.3% of variance in liking, 32.5% of variance in purchase intention, and 35.7% of variance in purchase; in these cases, the R^2^ values are relatively close to the moderate level. In addition, trust and influencer’s credibility and informative value explain 22.4% of variance in knowledge and 22.8% of variance in preference; therefore, in these two cases, the explanatory power of the model is relatively weak. With respect to the relationship between informative value and its effect on trust and influencer’s credibility, R^2^ reaches a value close to a moderate level (43.1%).

With regard to the effect size (f^2^), its evaluation has been carried out, taking into consideration the cutoffs points (0.02 = small; 0.15 = medium; 0.35 = large) suggested by Cohen (1988) [36]. The results show that the effect size of informative value is small for all consumers’ attitude phases, except for awareness, where it has an effect size close to the medium level (f^2^ = 0.127). Regarding trust and influencer’s credibility, the effect sizes reach levels higher than those corresponding to informative value, ranging from medium to low levels. The most important of these are the ones for awareness (f^2^ = 0.152) and purchase intention (f^2^ = 0.157). Finally, it is worth noting the high effect size of informative value on trust and influencer’s credibility (f^2^ = 0.757).

As far as the predictive relevance is concerned, all Q^2^ reach a positive value, which confirms the predictive performance of the structural model.

Regarding hypothesis 1, it can be asserted that the construct that combines trust and influencer’s credibility positively affects awareness, knowledge, liking, preference, purchase intention, and purchase. Therefore, in accepting hypothesis 1, we conclude that trust and influencer’s credibility positively affects consumers’ attitude phases, although their influence is stronger over purchase intention, awareness, and purchase.

Moving onto hypothesis 2, construct informative value positively affects consumer’ attitude phases, although its influence seems to be slightly lower than the one exerted by trust and influencer’s credibility. In this case, the effect over awareness is similar, but the rest are all lower. However, informative value also has an impact over the attitude phases by means of an indirect effect through trust and influencer’s credibility.

When testing hypothesis 3, the positive effect that informative value has over trust and influencer’s credibility is the strongest one that we discovered, in line with the findings of Lou and Yuan (2018). This is enough evidence to assert that informative value contributes to the construct trust and influencer’s credibility positively.

In the end, as predicted, followers’ attitude phases will be more influenced by the content if they perceive that it is relevant and trustable, as well as if they believe that the social media influencer has a high level of credibility. The link is stronger in the case of trust and influencer’s credibility than in informative value; however, the latter has an indirect effect on the attitude phases through the former, which was an interesting finding also predicted by Lou and Yuan (2018). Therefore, we can assert that informative value and trust and influencer credibility play a key role in determining to what extent exposure to the content will affect each of the attitude phases of the followers.

## 5. Discussion

The massive spreading in the use of SM changes the way people communicate, obtain, share, and process information for making all kinds of decisions [17,18]. The ability to assess and understand this relevance is key for both academics and practitioners to gain a better understanding of SM and its relationship with consumer behaviour, as well as businesses environments overall [19,24,37].

In this research, the focus is on the role of SM in motivating healthism by increasing health consciousness through instructional materials about healthy food properties, shared by nutritionists online, who occasionally become social media influencers. The results suggest that consumers find and use for their decision-making processes accessible and free information that is understandable, relevant, and credible [19,24]. The new information, incorporated into their decision-making processes, will alter the cognitive, affective, and behavioural attitude phases. Moreover, the intensity with which this happens will depend on the perceived informative value and trust in the influencer, as well as the influencer’s credibility, as perceived by the follower [20,21].

More specifically, the results obtained in this study suggest that the greatest influence is produced in the cognitive (awareness) and behavioural (purchase intention and purchase) stages, and not so much in the affective stage, with the effect of trust in the influencer generated content and credibility of the influencer being more relevant than the actual informative value of the shared content, relating to real food. However, it should also be noticed that trust in the content shared and influencer’s own credibility depends, to a large extent, on whether the content shared through SM is newsworthy for the followers.

Therefore, health professional influencers who aim to improve people’s quality of life by promoting healthier lifestyles must deal with the great challenge of achieving high levels of trust and credibility by providing their followers with content of differential value in the face of the enormous amount of messages about food that these followers receive every day through different channels.

It is also very interesting to note that the greatest influence occurs at the level of becoming aware of the importance of healthy eating habits and existence of healthy food alternatives to ultra-processed foods, as well as of the purchase intention and consumption of them, but not so much at the level of clearly preferring them to ultra-processed foods or adopting a clear critical view of them (liking). These findings promote an interesting line of research to understand why this happens. Some factors that could be taken into account are the taste of the products and consumption situations, as well as their smell, appearance, price, and even image.

The study has been developed in Spain, but it can be observed that the message conveyed by the influencer is slowly reaching international markets. This may be because the real food trend covers a hot topic in many Western countries, mainly the health problems associated with a change in diet towards more ultra-processed food [38,39,40,41,42]. Most of the time, the information shared by different bodies is confusing and even contradictory, which brings about confusion among consumers. In a moment when they feel that they should be making some changes to their diet, they have no idea whatsoever about where to start. This situation repeats itself in the most economically powerful regions, such as the United States or Europe.

The possibility of receiving constant updates about health-related topics by health professionals through SM is highly convenient for most people nowadays, and it has the potential of reaching large numbers of people worldwide [7]. Thus, SM becomes a great channel through which consumers can increase their health consciousness and implement healthier lifestyles.

Nevertheless, during our research, we noticed that many companies were not properly acquiring knowledge from non-branded content on SM. One such example is a firm that produces “gazpachos” and “salmorejos”, the two typical Spanish cold tomato soups. During a period, they experienced a notorious increase in their sales. The main reason was that the social media influencer participating in our study shared the “gazpachos” and “salmorejos” of that particular brand because they were the only ones that included virgin olive oil in the composition, which is highly considered by real food guidelines, in terms of health and nutrition. From the moment that he shared the product, followers wrote him complaining about the lack of supply of the product, that was usually out of stock, probably because the company did not expect the rapid growth in demand.

At some point, the company designed responses to the rise in demand. Unfortunately, they probably invested in increasing the capacity to be able to produce more but, at the same time, they decided to lower the cost of production by replacing the olive oil with sunflower oil, therefore, completely ignoring their genuine competitive advantage. From the moment the new version of the product landed in the shelves, several followers noticed this change in ingredients and shared the information with the influencer who, then, posted some content alerting about the modified version of the product to all his followers. The shelves are now almost always full, and it is not a product that he has continued to recommend or share with his followers.

In this particular case, the company could have realized that the key to the success was the use of olive oil and probably redesign their package to highlight this fact, at the same time as the production capabilities were improved to supply more products, and probably even a price increase, due to differentiation, could have been negotiated with the supermarket chain.

This example portrays the importance of knowledge acquired through SM in the food industry, especially now that people rely on these platforms to gain information and make decisions as consumers. Therefore, companies are not still giving it the importance that it deserves, and neither are they investing in understanding the whole impact the SM may have in their companies’ performance. That is the reason why we included four questions at the end of the survey, for respondents to give their opinions about situations similar to the one that has just been explained. Results show that 56% of respondents experienced shortage when looking for products that the nutritionist recommended on SM. Moreover, 83% of respondents think that the supply of products that can be considered real food or good processed food products is smaller than the supply of ultra-processed food, and 93% strongly agreed with the fact that food companies should invest in creating alternatives against ultra-processed food that would make it easier to adopt healthier lifestyles.

Considering these results, we encourage companies to listen to both the influencers and followers, include and give them active roles in the design of their products, and create solutions for this group of consumers, which is growing exponentially, especially among young people [41]. Therefore, a wide range of possibilities exist for companies to listen to food-related trends online and design suitable alternatives for them. In addition, the results of this study highlight the importance of designing informative contents that are understandable, credible, and relevant to the general public, which implies that companies should pay special attention to the design of their advertisements and labelling of their food products. Finally, companies could even get involved in the design of educational activities aimed at consumers, with the objective of raising awareness and stimulating them, not only at a behavioural level but also at a cognitive and affective level. This means acquiring with them, and with society as a whole, a real commitment to health through healthy food.

## 6. Conclusions

In this research, the holistic analysis of SM stimuli and its impact on consumer behaviour, using a more complex theory than the one chosen by previous literature, allow us to share implications for both academics and practitioner, in the hope that it will help understand how SM impacts consumers by increasing their health consciousness and changing their attitudes towards food products.

Food-related institutions can learn from the findings of this study and benefit from SM, in order to formulate strategies directed towards motivating healthy lifestyles and decreasing the incidence of chronic diseases associated with the diet [43]. When designing and implementing such strategies, it should be considered the influence of information accessible through SM on consumers’ decisions, at the cognitive, affective, attitudinal, and behavioral levels. This influence seems to be stronger in the early stages of the attitude phase model. This may be because consumers will intuitively incorporate the new information shared by the influencer on SM that is free, easy, and enticing. However, for actually preferring or buying a product, more engagement is required from the consumer; therefore, more variables regulating this link will come into play. This shift initially requires a change from habitual buying behaviour, in which involvement is very low because consumers have been purchasing food as a commodity for a long time, to complex buying behaviour, which requires a higher level of involvement that may include activities such as learning all the differences between food-related products, using the knowledge garnered on SM.

In addition, special priority should be given to the transmission of understandable and useful content, provided by credible and trustworthy individuals and institutions through responsible, coherent, and coordinated strategies and actions.

Furthermore, there is an interest in incorporating into food research variables to identify differences between consumers, in terms of their motivation, interest, and commitment to healthy food. For instance, our research reveals a clearly higher follow-up by women and the younger age groups, with higher levels of education and middle-income levels.

In the specific case of our study context (Spain), several official reports have detected an increase in the consumption of ultra-processed meat products, compared to a decrease in the consumption of fresh or minimally processed meat products [44,45]. Hence, the importance of placing value on strategies (in our case, social networks and expert professional influencers) that can contribute to the idea that natural food and well-processed foods (real food) are compatible with any food category, including quality meat products.

Finally, further research can apply the framework provided in this research to analyse the effect of SM in consumer behaviour, related to food in other countries, to test the significance of these findings worldwide.

## Figures and Tables

**Figure 1 foods-11-00224-f001:**
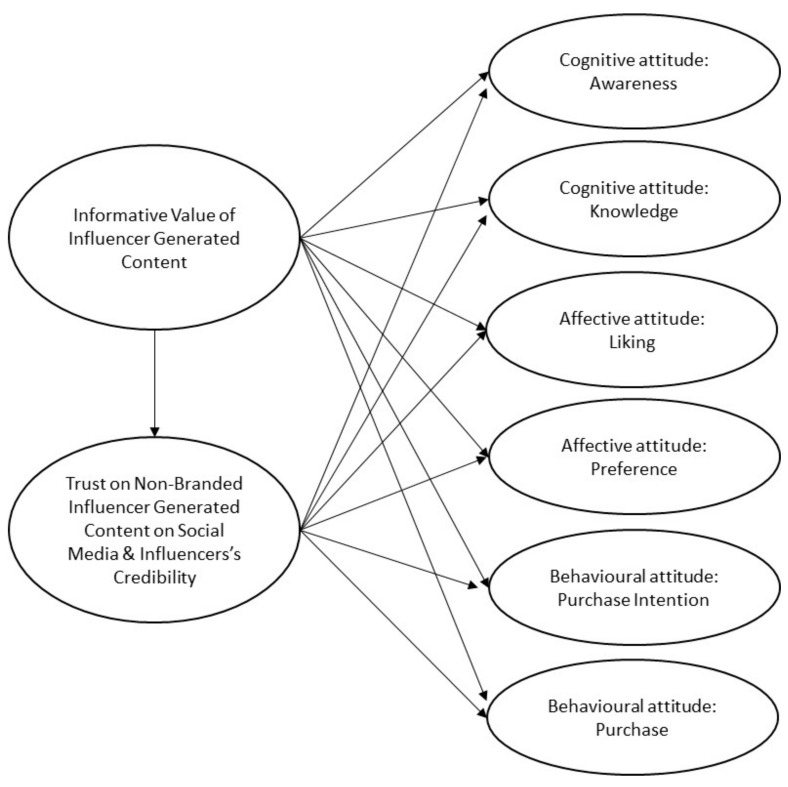
Theoretical model.

**Figure 2 foods-11-00224-f002:**
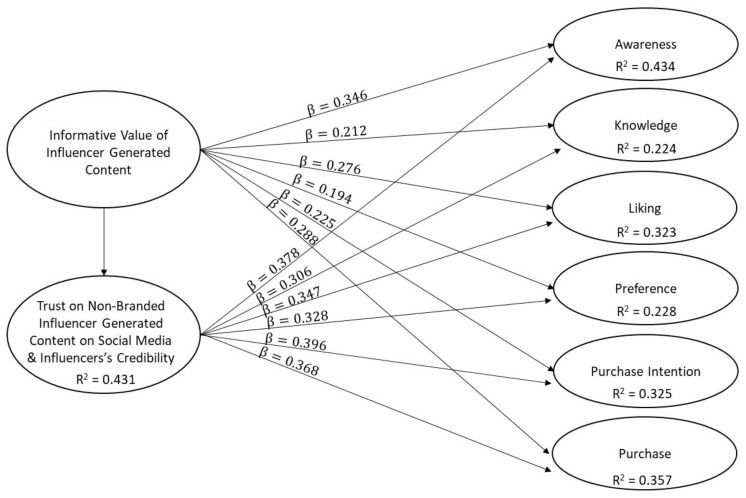
Partial least squares (PLS) path model.

**Table 1 foods-11-00224-t001:** Socio-demographic and consumer-related characteristics of the sample.

Variable	Percentage%
**Gender**	
Female	87.8
Male	11.9
**Age**	
Below 18	4
From 19 to 25	34
From 26 to 35	40
From 36 to 45	15
From 46 to 55	6
From 56 to 65	1
66 or above	0
**Nationality**	
Spain	95.8
Other	4.2
**Level of Education**	
Primary education or below	0.7
Secondary education (first stage)	5.2
Secondary education (second stage)	11.3
Higher education	82.7
**Income Level/Social Strata**	
Low	3.9
Medium-low	20.9
Medium	61.5
Medium-high	13
High	0.7

The socio-demographic factors suggest that the respondents were mainly Spanish young females, with a high level of education (82.7%) and perception of belonging to the medium social strata, meaning a moderate amount of income within the context of Spain.

**Table 2 foods-11-00224-t002:** Measurement model assessment.

Construct	Indicator ^a^	Weight	Loadings	AVE ^b^	Composite Reliability
Informative Value (IV)	IV1	1.000	1.000	1.000	1.000
Trust (T) + Influencers’ Credibility (IC)	T1	0.180	0.803	0.613	0.917
	T2	0.182	0.816		
	T3	0.174	0.779		
	T4	0.192	0.858		
	IC1	0.180	0.804		
	IC2	0.176	0.787		
	IC3	0.137	0.711		
Awareness (AW)	AW1	0.308	0.689	0.503	0.801
	AW2	0.325	0.728		
	AW3	0.356	0.798		
	AW4	0.272	0.709		
Knowledge (KN)	KN1	1.000	1.000	1.000	1.000
Liking (LK)	LK1	0.579	0.816	0.615	0.761
	LK2	0.534	0.751		
Preference (PR)	PR1	0.528	0.855	0.750	0.857
	PR2	0.541	0.877		
Purchase Intention (PI)	PI1	0.684	0.730	0.513	0.780
	PI2	0.508	0.662		
Purchase (P)	P1	0.269	0.694	0.598	0.855
	P2	0.265	0.682		
	P3	0.332	0.855		
	P4	0.329	0.846		

^a^ For the full meaning and wording of each indicator, please refer to Appendix A. ^b^ Average Variance Extracted (AVE).

**Table 3 foods-11-00224-t003:** Discriminant validity (Fornell and Larker ^a^).

	A	IV	K	L	P	Pch	PI	TIC
Awareness (A)	0.709							
Informative Value (IV)	0.594	1.000						
Knowledge (K)	0.771	0.413	1.000					
Liking (L)	0.760	0.504	0.765	0.784				
Preference (P)	0.818	0.409	0.604	0.812	0.866			
Purchase (Pch)	0.740	0.529	0.549	0.708	0.695	0.774		
Purchase Intention (PI)	0.774	0.485	0.530	0.685	0.687	0.809	0.843	
Trust + Influencers’ credibility (TIC)	0.605	0.656	0.445	0.528	0.455	0.556	0.544	0.783

^a^ Diagonal values (square root of AVE) should be higher than of-diagonal values (correlations).

**Table 4 foods-11-00224-t004:** Discriminant validity (HTMT ^a^ ratio).

	A	IV	K	L	P	Pch	PI	TIC
Awareness (A)								
Informative Value (IV)	0.603							
Knowledge (K)	0.770	0.413						
Liking (L)	0.831	0.504	0.765					
Preference (P)	0.816	0.409	0.604	0.814				
Purchase (Pch)	0.848	0.525	0.542	0.698	0.675			
Purchase Intention (PI)	0.806	0.487	0.535	0.706	0.707	0.815		
TIC	0.616	0.658	0.447	0.531	0.456	0.555	0.549	

^a^ Threshold value should be below 0.9.

**Table 5 foods-11-00224-t005:** Structural path estimates.

Relationship	Path Estimates	Standard Deviation	T-Statistics	*p* Values
Informative value→Awareness	0.346	0.026	13.446	0.000
Informative value→Knowledge	0.212	0.021	10.180	0.000
Informative value→Liking	0.276	0.026	10.539	0.000
Informative value→Preference	0.194	0.027	7.213	0.000
Informative value→Purchase	0.288	0.022	12.962	0.000
Informative value→Purchase Intention	0.225	0.029	7.863	0.000
Informative value→Trust and Influencers’ Credibility	0.656	0.016	41.785	0.000
Trust and Influencers’ Credibility→Awareness	0.378	0.028	13.341	0.000
Trust and Influencers’ Credibility→Knowledge	0.306	0.025	12.408	0.000
Trust and Influencers’ Credibility→Liking	0.347	0.029	12.115	0.000
Trust and Influencers’ Credibility→Preference	0.328	0.027	12.019	0.000
Trust and Influencers’ Credibility→Purchase	0.368	0.024	15.313	0.000
Trust and Influencers’ Credibility→Purchase Intention	0.396	0.031	12.833	0.000

**Table 6 foods-11-00224-t006:** Indirect effects path estimates.

Indirect Effects	Path Estimates	St Deviation	T-Statistics	*p* Values
Informative Value→Trust and Influencers’ Credibility→Awareness	0.248	0.020	12.577	0.000
Informative Value→Trust and Influencers’ Credibility→Knowledge	0.201	0.018	11.299	0.000
Informative Value→Trust and Influencers’ Credibility→Liking	0.228	0.020	11.456	0.000
Informative Value→Trust and Influencers’ Credibility→Preference	0.215	0.019	11.237	0.000
Informative Value→Trust and Influencers’ Credibility→Purchase	0.241	0.017	14.117	0.000
Informative Value→Trust and Influencers’ Credibility→Purchase Intention	0.260	0.021	12.124	0.000

**Table 7 foods-11-00224-t007:** Effect size f^2^.

	A	K	L	P	Pch	PI	TIC
Informative Value (IV)	0.127	0.033	0.064	0.028	0.073	0.041	0.757
Trust and Influencers’ Credibility (TIC)	0.152	0.069	0.101	0.079	0.119	0.157	

Awareness (A); knowledge (K); liking (L); preference (P); purchase (Pch); purchase Intention (PI).

## Data Availability

Data available on request.

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
