# Peer review of "Promoting Sustainable Lifestyle Habits: “Real Food” and Social Media in Spain"

_foods, 2022, doi:10.3390/foods11020224_

Round 1

Reviewer 1 Report

The paper is about the influence of social media on consumer’s attitudes. It is a fascinating study. I’ve read with interest. I’ve made some suggestions, I hope you find them helpful.

I don’t think the paper is adequate for the special issue “Open Innovation in Meat and Meat Products”. The authors should select another special issue (or none), in my opinion.

Title- It is clear to me

Abstract – The first section of the abstract is very extended and generic. The authors could better use the abstract space to describe the methods and results clearly.

Introduction –

L31 - typo

L67 – I found very awkward the term “real food”. Do the nutritionists in Spain really use this term? Do you have a reference for this?

 Food processing is the base of some food guides (like Brazil's). Does the Spain food guide also describe food processing?

I think you should include some references for the food processing (e.g. Monteiro and colleagues)

L74 – which profile? I’m confused here.

L75 he? Who is “he”?

L74 – 82 – This paragraph is tough to follow. Is it a result section?

L89 – His? Why do you are using sex pronouns for an Instagram profile?

You use too many acronyms. Some like “real fooding” (RF) it is not necessary. Use only for terms that is well known for their acronym (like WHO for ex.)

Figure 1 – The model is not correct. I know you’ve done this to simplify, but you have to insert each effect arrow from the informative value and trust on all other latent variables.

Why do you have light and dark grey circles? What is the difference?

Methods

L228 – This should be in the introduction

Topic 3.2. Too many acronyms. Sorry, but I had to write on paper all acronyms to understand the text.

Data analysis must be greatly improved.

Please include thresholds for factor loadings, AVE, composite reliability.

I presume you used bootstrapping. How many samples were employed?

How did you measure multicollinearity? How did you check the predictive relevance?

Results – I found the results hard to follow. The results are repeated in the tables, figure and text. Please choose only one and be consistent. I suggest reading other studies using PLS-SEM. I’ve missed the VIF analysis.

Please format the tables adequately. They have extra spacing.

L381-385 – This is discussion, not result.

Why use composite reliability and cronbach’s alpha since both measure reliability? I suggest CR due to alpha’s tau-equivalence assumption.

Table 4 – please include in the title that you measured using Fornell Larcker’s criteria (there are others like HTMT – which I personally prefer).

Table 5 and 6 – please include an arrow indicating the effect between the variables

Figure 2 has a very low resolution. Include in the figure the letter “beta” before the values and R² inside the saquares.

Table 5 can be removed as this figure better depicts the result.

Please include and discuss the effect size (f ²) of each effect.

L445 – 453 You do not have to repeat all the results, they're depicted in the figure.

The further sections must be presented with the effect size. A result can be significant but with a small effect size.

I think the study could benefit from some professional English proofreading services.

Author Response

Comments and Suggestions for Authors

Point 1: The paper is about the influence of social media on consumer’s attitudes. It is a fascinating study. I’ve read with interest. I’ve made some suggestions, I hope you find them helpful.

Response 1: First of all, we would like to thank the reviewer for his or her interest and positive evaluations and encouragement, as well as for his or her detailed review and valuable suggestions and indications, which undoubtedly allow us to increase the contribution of this work to this field of knowledge.

Point 2: I don’t think the paper is adequate for the special issue “Open Innovation in Meat and Meat Products”. The authors should select another special issue (or none), in my opinion.

Response 2: One of the food categories that most needs to be associated with Real Fooding is meat foods, given the high number of ultra-processed meat products currently being sold and consumed. This issue has been expressly referred to in the text of the paper. However, thanks to your comment, we have expanded on this issue in both the Introduction and Conclusions sections. More precisely, certain tendencies have been identified among Spanish consumers to associate meat products with unhealthy foods, undervaluing their beneficial health properties. However, at the same time, they are increasing their consumption of ultra-processed meat products (Consumption Reports of the Spanish Ministry of Agriculture, Fisheries and Food, 2018, 2020). Hence the importance of highlighting strategies (in our case, social networks and expert professional influencers) that can contribute to the idea that natural food and good-processed food (real food) is compatible with any food category, including quality meat products. The importance of this trend in the context of our study has led the editors of this special issue to make us a proposal for a contribution to it.

https://www.mapa.gob.es/images/es/20190807_informedeconsumo2018pdf_tcm30-512256.pdf

https://www.mapa.gob.es/es/alimentacion/temas/consumo-tendencias/informe-anual-consumo-2020-v2-nov2021-baja-res_tcm30-562704.pdf

Title- It is clear to me.

Point 3: Abstract – The first section of the abstract is very extended and generic. The authors could better use the abstract space to describe the methods and results clearly.

Response 3: Following the reviewer's indications, the abstract has been modified to reduce the first part, with a more generic content, and the information on methods and results has been expanded, respecting the maximum length allowed.

Introduction –

Point 4: L31 – typo

Response 4: The mistake has been corrected.

Point 5: L67 – I found very awkward the term “real food”. Do the nutritionists in Spain really use this term? Do you have a reference for this?

Response 5: Indeed, "Real Food" is a term employed in the context of nutrition both in Spain and in other countries. In Spain, the English term "real food" is used as well as the Spanish term "comida real", both in specialized areas and in the general media. The use of the terminology can be consulted, for instance, in these links:

https://projekter.aau.dk/projekter/files/334476154/The_influence_of_Realfooding_on_consumer_behaviour_in_Spain_10th_Semester_Master_Thesis_.pdf

https://d1wqtxts1xzle7.cloudfront.net/65040919/Movimientos_Sociales_en_la_era_digital_el_caso_realfooding_y_el_impacto_en_los_supermercados-with-cover-page-v2.pdf?Expires=1641031701&Signature=f9Xt~qnD4y419Gr35mJZRgHC-4ZcFgQ8ygWv3zrZ6IauAtU3p0-WUfyHakQpQJL0N9dGc9v3X35wThsl-A2c4X-1krjYlzAzN9XzYoI29lxwUZO36JuGCpFvEBv~pIvCdmrniq7yoeMG2GWzMedomfMyvonMcIq3uFYPIEwkTUaPSxNOK1lcbR6EsFhBpDxU7EMRqhAamGV4n3ZHkh0DM1YpAspAJa87F8JKRuxjoxIOgxHiWxXgKKJCounUMQuqP8A-YTFHo49aVhFz9tZ35OapFcITOOoFrVuqwkTs3fwLmIVvE3XA1QrgbP~yWUfcGG~DDl7cyu1UaE2csD5nXw__&Key-Pair-Id=APKAJLOHF5GGSLRBV4ZA

https://revistascientificas.us.es/index.php/Ambitos/article/download/11011/10173 (p. 102-122)

https://realfooding.com/

https://elcomidista.elpais.com/elcomidista/2019/07/09/articulo/1562702626_450013.html

https://www.lne.es/vida-y-estilo/2021/08/22/real-food-escuchas-hablar-19002951.html

https://www.elespanol.com/ciencia/nutricion/20190420/nutricionista-carlos-rios-advierte-no-evidencias-saludable/391961621_0.html

https://www.thefork.es/blog/real-food-que-es

https://www.cinib.es/blog/132-nutricion/311-real-food-la-tendencia-que-arrasa-en-internet

https://www.sciencedirect.com/science/article/pii/S0260877407001045?casa_token=xXUcItK5o-wAAAAA:Vo0vJpPIli_JPi0IZIiagFkeK2ugwvsP8xYKnGeepBNfHcNaoKNZf954E1DTD_vRskFaSpNn1PQ

https://uvadoc.uva.es/handle/10324/42364

https://buleria.unileon.es/handle/10612/11418

https://www.thefork.es/blog/real-food-que-es

Campillo, S. “La ciencia detrás del movimiento real food, ¿está avalado por estudios o es solo una moda?”. Vitónica. Retrieved from: https://www.vitonica.com/alimentos/la-ciencia-detras-del-movimiento-real-food-esta-avalado-por-estudios-o-es-solo-una-moda (2018, May 14th).

Carpallo, S. “Qué debes saber de la ‘comida real’, la dieta de la que todos hablan”. El País, SModa. Retrieved from: https://smoda.elpais.com/belleza/que-debes-saber-de-la-comida-real-la-dieta-de-la-que-todos-hablan/ (2018, August 14th).

Díaz Madurga, L. “Realfooding: volver a comer como lo hacían tus abuelos”. Gastroactitud. Retrieved from: https://www.gastroactitud.com/pista/realfooding-volver-a-comer-como-lo-hacian-tus-abuelos/ (2019; April 30th).

Heidemeyer, P. “¿Qué es el “realfooding”? Bezzia. Retrieved from: https://www.bezzia.com/que-es-el-realfooding/ (2019, July 21th).

Ruiz De La Prada, S. “'Realfooding': el movimiento que arrasa en Internet y que debes conocer”. Hapers’ Bazaar. Retrieved from: https://www.harpersbazaar.com/es/belleza/dieta-ejercicios-adelgazar-belleza/a23304101/dieta-comida-realfooding-movimiento-comida-real-que-es/ (2018, October 15th).

Ríos, C. (2019). Come comida real: Una guía para transformar tu alimentación y tu salud. Madrid: Paidós.

Point 6: Food processing is the base of some food guides (like Brazil's). Does the Spain food guide also describe food processing?

Response 6: Spain does not have its own guide to define processed and ultra-processed foods. It takes several systems as a reference:

- NOVA (Public Health School, Sao Paulo, Brazil) system.

- SIGA system (France)

- Regulation (EC) No. 852/2004

Detailed information can be found in the following links, which provides access to the Report of the Scientific Committee of the Spanish Agency for Food Safety and Nutrition (AESAN) on the Impact of consumption of ultra-processed foods on the health of consumers:

Comité Científico AESAN. (Grupo de Trabajo) Talens, P., Cámara, M., Daschner, A., López, E., Marín, S., Martínez, J.A. y Morales, F.J. Informe del Comité Científico de la Agencia Española de Seguridad Alimentaria y Nutrición (AESAN) sobre el impacto del consumo de alimentos “ultra-procesados” en la salud de los consumidores. Revista del Comité Científico de la AESAN, 2020, 31, pp: 49-76.

https://www.aesan.gob.es/AECOSAN/docs/documentos/seguridad_alimentaria/evaluacion_riesgos/informes_cc_ingles/ULTRA-PROCESSED_FOODS.PDF

https://www.aesan.gob.es/AECOSAN/docs/documentos/seguridad_alimentaria/evaluacion_riesgos/informes_comite/ULTRAPROCESADOS.pdf

Point 7: I think you should include some references for the food processing (e.g. Monteiro and colleagues).

Response 7: Thank you very much for the suggestion. The following food processing references have been included:

Monteiro, C.A.; Cannon, G.; Levy, R.B;, Moubarac, J.C.; Louzada, M.L.C.; Rauber, F.; Khandpur, N.; Cediel, G.; Neri, D.; Martinez-Steele, E.; Baraldi, L.G.; Jaime, P.C. Ultra-processed foods: What they are and how to identify them. Public Health Nutrition 2019, 22, 936-941.

Monteiro, C.A.; Moubarac, J.C.; Levy, R.B.; Canella, D.S.; Louzada, M.L.; Cannon, G. Household availability of ultra-processed foods and obesity in nineteen European countries. Public Health Nutrition 2017, 21, 18-26.

Talens, P; Cámara, M.; Daschner, A.; López, E.; Marín, S.; Martínez, J.A.; Morales, F.J. Informe del Comité Científico de la Agencia Española de Seguridad Alimentaria y Nutrición (AESAN) sobre el impacto del consumo de alimentos “ultra-procesados” en la salud de los consumidores. Revista del Comité Científico de la AESAN, 2020, 31, pp: 49-76.

Point 8: L74 – which profile? I’m confused here.

Point 9: L75 he? Who is “he”?

Point 11: L89 – His? Why do you are using sex pronouns for an Instagram profile?

Response 8, 9 and 11: Thank you for your feedback. To clarify these questions regarding the Instagram profile of the influencer who collaborated in our research and the content of this section, we have incorporated information related to the specific case study. To this end, content has been moved from the "Case Study" section to the "Introduction" section, as indicated by the reviewer. In addition, it has been clarified that the nutritionist who collaborated with us in the research is a man.

Point 10: L74 – 82 – This paragraph is tough to follow. Is it a result section?

Response 10: A correction has been made in the paragraph indicated to clarify its content. It is an observation derived from the evolution of the number of followers of the nutritionist under consideration, from the papers reviewed in this study, and from the experience that the nutritionist himself has reported to us. Our study starts from these premises to really check if they are valid, based on the information provided by the followers.

Point 11: L89 – His? Why do you are using sex pronouns for an Instagram profile?

Response 11: This point has been answered previously, together with points 8 and 9.

Point 12: You use too many acronyms. Some like “real fooding” (RF) it is not necessary. Use only for terms that is well known for their acronym (like WHO for ex.)

Response 12: Thank you for the indications. All acronyms have been removed, except for SM (Social Media), given their frequent and widespread use. Appendix A, which contained the list of acronyms initially used, has been removed.

Point 13: Figure 1 – The model is not correct. I know you’ve done this to simplify, but you have to insert each effect arrow from the informative value and trust on all other latent variables.

Point 14: Why do you have light and dark grey circles? What is the difference?

Response 13 and 14: Figure 1 has been modified according to the reviewer's indications. To clarify the image, the grey color, which was simply for aesthetic reasons, has been eliminated.

Methods

Point 15: L228 – This should be in the introduction

Response 15: Following the reviewer's indications, a part of the information related to "case selection" has been included in the Introduction section.

Point 16: Topic 3.2. Too many acronyms. Sorry, but I had to write on paper all acronyms to understand the text.

Response 16: All acronyms, except SM, have been removed to improve the comprehension of the text.

Data analysis

Point 17: Please include thresholds for factor loadings, AVE, composite reliability.

Response 17: Information on thresholds for factor loadings, AVE and composite reliability has been incorporated in the text.

Point 18: I presume you used bootstrapping. How many samples were employed?

Response 18: Following the recommendations of Hair et al. (2019), we have used 5000 resamples in the bootstrapping process (this has been specified in the text).

Point 19: How did you measure multicollinearity? How did you check the predictive relevance?

Response 19: To measure multicollinearity, VIF values were used. According to the VIF values, they all range from 1.575 to 2.740. Therefore, no VIF value exceeds the suggested threshold value of 3.0, which implies that we have not encountered collinearity problems. The predictive power was tested by means of the Stone-Geisser’s Q2 values, all of which have a positive value. All the information described above has been incorporated into the text of the paper.

Results

Point 20: I found the results hard to follow. The results are repeated in the tables, figure and text. Please choose only one and be consistent. I suggest reading other studies using PLS-SEM.

Response 20: Following the reviewer's indications, several modifications have been made. Specifically, table 2, which contains the information relating to Descriptive analysis of the study variables, has been integrated into Appendix B, and the rest of the tables have been renumbered. Information redundant with that presented in the different tables of results, as well as in figure 2, has also been removed from the text.

Point 21: Please format the tables adequately. They have extra spacing.

Response 21: The format of the tables has been revised according to the guidelines of the journal.

Point 22: L381-385 – This is discussion, not result.

Response 22: The information contained in the above lines has been transferred to the discussion and conclusion sections.

Point 23: Why use composite reliability and cronbach’s alpha since both measure reliability? I suggest CR due to alpha’s tau-equivalence assumption.

Response 23: Following the reviewer's indications, the information relating to cronbach's alpha has been removed, and that relating to composite reliability has been maintained.

Point 24: Table 4 – please include in the title that you measured using Fornell Larcker’s criteria (there are others like HTMT – which I personally prefer).

Response 24: Following the reviewer's indications, the reference to the Fornell Larcker's criteria has been included in table 3 (previously table 4). In addition, the information concerning HTMT has been incorporated (table 4).

Point 25: Table 5 and 6 – please include an arrow indicating the effect between the variables.

Response 25: Arrows indicating the effect between variables have been included in tables 5 and 6.

Point 26: Figure 2 has a very low resolution. Include in the figure the letter “beta” before the values and R² inside the saquares.

Response 26: Figure 2 has been modified according to the reviewer's indications: resolution has been improved, and letter “beta”and R² have been included.

Point 27: Please include and discuss the effect size (f ²) of each effect.

Response 27: Once the predictive power of the model has been analysed (R2), the effect size (f ²) has been assessed; the cutoffs points suggested by Cohen (1988) have been taken as a reference in the analysis.

Point 28: L445 – 453 You do not have to repeat all the results, they're depicted in the figure.

Response 28: Taking into account this suggestion, the above paragraph has been revised, simplifying its writing at the numerical level, thus avoiding reiterating the information contained in figure 2.

Point 29: The further sections must be presented with the effect size. A result can be significant but with a small effect size.

Response 29: Comments have been incorporated in the following sections concerning the conclusions of the effect size analysis.

Point 30: I think the study could benefit from some professional English proofreading services.

Response 30: The text has been edited by a native English-speaking expert.

Reviewer 2 Report

29-DEc-2021

Reviewer's Comments to Authors:

Recommendation: Minor Revision

Manuscript Foods-1528321 with title " Promoting sustainable lifestyle habits: "Real Food" and Social Media in Spain " provides an interesting research about health professionals are increasingly using social media to connect with the public. Study investigates how exposure to content created by health professionals, and posted on social media, influences consumers' attitudes towards recommended products.

I have the following comments:

ABSTRACT

  1. The abstract should state the study period, study type and sampling technique.
  2. Please clarify more the purpose of the study.
  3. Please revise your findings and originality of the research.

INTRODUCTION AND METHODS

Please clarify: At the same time, consumers are also confronted with nutrition and health claims (NHC) shared by companies in the marketing campaigns of their products, from commercials to packaging. NHC are claims in any message conveyed in text or image that state, suggest or imply that a food has beneficial nutrition properties."

Isn't the placement of NHC on food products controlled by law?

RESULTS and DISCUSSION:

There are too many tables and figures. Perhaps consider linking them more clearly or showing some of them (descriptive analysis) also in Appendix.  Please define statistical methods in the footnote and the content of the tables should be such that the data are of sufficient resolution for comfortable reading.

LITERATURE

The manuscript shows the relevant references related to this study.

CONCLUSION

Please write conclusion more clearly. Revise your findings and originality of the research and should provide a clear scientific justification for the research. The most important findings (results) should be more emphasized.

Author Response

Reviewer's Comments to Authors:

Recommendation: Minor Revision

Point 1: Manuscript Foods-1528321 with title " Promoting sustainable lifestyle habits: "Real Food" and Social Media in Spain " provides an interesting research about health professionals are increasingly using social media to connect with the public. Study investigates how exposure to content created by health professionals, and posted on social media, influences consumers' attitudes towards recommended products.

Response 1: First of all, we would like to thank the reviewer for his or her interest and positive evaluations and encouragement, as well as for his or her detailed review and valuable suggestions and indications, which undoubtedly allow us to increase the contribution of this work to this field of knowledge.

I have the following comments:

Point 2: ABSTRACT

  1. The abstract should state the study period, study type and sampling technique.
  2. Please clarify more the purpose of the study.
  3. Please revise your findings and originality of the research.

Response 2: Following the reviewer's indications, the abstract has been modified to reduce the first part, with a more generic content, and the information on methods, purpose, study period and findings has been expanded, respecting the maximum length allowed.

Point 3: INTRODUCTION AND METHODS

Please clarify: At the same time, consumers are also confronted with nutrition and health claims (NHC) shared by companies in the marketing campaigns of their products, from commercials to packaging. NHC are claims in any message conveyed in text or image that state, suggest or imply that a food has beneficial nutrition properties."

Isn't the placement of NHC on food products controlled by law?

Response 3: Indeed, in Spain there are official regulations both for the content of the labelling of food products and for the communication campaigns linked to them. However, it is also true that consumers often experience difficulties in understanding this information, sometimes because of its technical nature, sometimes because of confusion over certain terminology (e.g. natural, healthy, organic, ecological, processed, ultra-processed, etc.), and sometimes because they receive messages that are sometimes confusing or even contradictory. In fact, at the moment, there is a general controversy about meat products in Spain.

https://www.aesan.gob.es/AECOSAN/web/seguridad_alimentaria/subdetalle/futura_legislacion.htm

https://www.eldiario.es/economia/alberto-garzon-pide-espanoles-coman-carne-perjudica-salud-planeta_1_8112232.html

Point 4: RESULTS and DISCUSSION: There are too many tables and figures. Perhaps consider linking them more clearly or showing some of them (descriptive analysis) also in Appendix. Please define statistical methods in the footnote and the content of the tables should be such that the data are of sufficient resolution for comfortable reading.

Response 4: Thank you very much for these remarks. With regard to the tables and figures, both the format and the content of the tables and figures have been clarified. The aim has been to make them easier to read and to reduce the space they take up in the text. In addition, certain clarifications have been incorporated into the text of the paper to make them easier to understand and to link them better. As suggested by the reviewer, the table containing the descriptive analysis has been transferred to Appendix B. Finally, in those tables containing methodological information, footnotes have been included to improve their interpretation.

Point 5: LITERATURE

The manuscript shows the relevant references related to this study.

Point 6: CONCLUSION

Please write conclusion more clearly. Revise your findings and originality of the research and should provide a clear scientific justification for the research. The most important findings (results) should be more emphasized.

Response 6: Following the reviewer's suggestions, both the Discussion and the Conclusion have been developed more extensively, so that the originality of the research, its usefulness, its main findings and, ultimately, its scientific justification are more clearly highlighted.

Round 2

Reviewer 1 Report

All my concerns were addressed.